# WHEN AI DESCRIBES RACE?
# UNVEILING RACIAL BIAS IN VISION-LANGUAGE MODELS IN BRAZILIAN PEOPLE

**Leodécio Braz**
Instituto de Computação
Universidade Estadual de Campinas (UNICAMP)
l230219@dac.unicamp.br

**Denise Carvalho**
Instituto de Artes
Universidade Estadual de Campinas (UNICAMP)
denisecs@unicamp.b

**Marcos M. Raimundo**
Instituto de Computação
Universidade Estadual de Campinas (UNICAMP)
mrai@unicamp.br

## ABSTRACT

Multi-modal AI Models that intrinsically reproduce undesirable social biases remain a critical challenge; racial diversity and fairness are key concerns in AI literature. This paper investigates racial bias in captioning models, focusing on how different racial groups are represented in generated text sentences. We explored a new dataset, TSE (a self-hetero-identified dataset from Brazil), and two others (UTKFace, FairFace) to analyze State-Of-The-Art models to assess their racial attribution disparities. Our findings reveal that certain racial groups are disproportionately referenced, with the white race often being treated as the default while other races receive explicit mentions at varying rates. This discrepancy reflects intrinsic societal biases embedded in widespread models, perpetuating racial stereotypes and reinforcing systemic inequities. We also reveal that other racial domains, such as Brazilian, are poorly captured by existing models, leading to disparities in racial representation. Our study underscores the urgent need for bias mitigation strategies in generative Image-Text-to-Text models, ensuring fairer and more inclusive representation across diverse racial identities.

## 1 INTRODUCTION

Artificial Intelligence (AI) models are now deeply embedded in daily life, assisting with tasks like content generation, identification, filtering, and classification Huang et al. (2022); Dwork et al. (2012). This widespread use means AI decisions directly impact people, making transparency and fairness crucial, thus rendering the "black box" concept obsolete. Moreover, it is concerning and notorious that algorithms may reproduce racial and gender disparities Obermeyer et al. (2019).

Since the advances in the AI area, many studies have demonstrated that AI algorithms can discriminate based on race, gender, or other individual characteristics Buolamwini & Gebru (2018). Racial bias is a key concern of Machine Learning (ML) models and is present when an algorithm systematically favors one race over another, propagating and perpetuating prejudices Huang et al. (2022). The emergence of Generative AI has amplified debates and concerns surrounding biases against specific groups based on race, gender, and other characteristics Park (2024). These biases often manifest through the reinforcement of stereotypes Bianchi et al. (2023), the propagation of social violence Gebru (2020); Heikkilä (2022) or reproducing data imbalances Currie et al. (2024).

A central concern in this discussion is how "Whiteness" is treated by these models. As noted by Park (2024), "Whiteness has been historically and systematically deemed as colorless and 'the human norm', which contributes to maintaining White normalcy and privileges." In this work, we want to reinforce this racial debate and bring more elements to contribute to this discussion about how White is often not interpreted as a racial concept, while non-White races are made explicit by AI models.

To study such a challenging topic, we introduce a pipeline to measure explicit racial references on generated texts statistically. The statistical measures come from the principle that to achieve equality, racial references should be similar across all groups without "privileging" only a particular group. We also discuss our findings for model behavior in the Brazilian racial context and highlight critical observations. Our main contributions are summarized as follows:

- Proposal of an automated pipeline for evaluating racial presence *vs* absence in captions generated by two State-Of-The-Art (SOTA) models.
- Assessment and discussion of captioning models performance in diverse racial contexts, specifically focusing on the Brazilian racial domain.
- Experiments conducted on two benchmark datasets reveal intrinsic racial biases in captioning models, highlighting their role in perpetuating stereotypes and other racial disparities.

## 2 CONTEXT AND RELATED WORK: THE RACE ISSUE IN BRAZIL AND COMPUTATIONAL SYSTEMS

The use of A.I. systems in decision-making processes has highlighted their potential to perpetuate and accentuate racial inequalities Wang et al. (2020); Deuschel et al. (2020). Machine learning models and generative systems, in particular, intrinsically encode biases that impact marginalized and minority groups Hendricks et al. (2018).

### 2.1 THE ISSUE OF RACE AND THE BRAZILIAN CONTEXT

One of the premises of this study is related to the concept of race as a socio-political and cultural construct rather than a scientific-biological category, as explained by Guimarães (2003). Beyond etymological roots, racism was developed on the perception of their interrelationship from the constructed meanings within the multiple dimensions of everyday social experiences.According to Mullings (2005), how racial constructs, i.e., the idea of race itself, are constructed, occurs in the most diverse geographical spaces based on repertoires of beliefs, symbols, and practices that, once transmitted from the past, are reinterpreted in subsequent historical moments. However, such racial classification has weak scientific foundations.

Brazilian racial culture has been historically marked by the belief that subordinate social positions are appropriate for people classified as Black (Negro) based on their phenotypic traits. According to Bento et al. (2002), the Brazilian racial context is permeated by phenomena in the social imaginary of invisibility and stigmatization that associate Black individuals with a negative legacy as a consequence of the slavery period and the persistence of inequalities, stigmas and discrimination resulting from this historical process, while the White population benefits from a range of privileges maintained under a structural spectrum Müller & Cardoso (2018). Schucman & Schlickmann (2018) point out that the maintenance of this structure favors the protection of these privileges, whether material or symbolic, as well as allowing the repeated enactment of these social and cultural logic in the trajectory of Brazilian society.

As part of the whitening process in Brazil, strategies of political and scientific narratives reinforced racial diversity as a positive aspect, as explained by Carrera (2024), making Pardos classified as White and Black being classified as Pardos Lima (2022). In Brazil, Pardo is a sociocultural term that describes individuals of mixed ancestry, often encompassing a diverse range of skin tones and racial identities Gomes (2019). This present study shows this whitening process in the AI models at a certain level, vanishing the cultural and ethnic characteristics of the Pardo group by frequently characterizing them as white. However, in the social studies about race in Brazil, White individuals generally appear as the universal model of humanity with its full privileges. In contrast, Pardo individuals do not benefit from this proximity of color, as they are not seen as white. This category poses unique challenges for generative models, which may struggle to accurately interpret or represent the nuanced racial identities within this group, leading to oversimplified or biased outputs.

When stereotypes and systemic inequalities manifest automatically or indirectly, they remain significant barriers to achieving equity within modern ethical and legal spheres. In the Brazilian context, this means that the historical legacy of 'whitening' and the erasure of mixed-race identities are not merely social relics; they are algorithmically encoded into the very models that shape our digital

reality. As argued by Ruback et al. (2022), identifying and critically examining these sociocultural foundations is a necessary prerequisite for understanding their downstream effects in AI systems and developing robust technical strategies for bias mitigation.

## 2.2    RACIAL BIAS IN MACHINE LEARNING AND GENERATIVE MODELS

Racial bias in machine learning has become a critical area of research, as biased algorithms can perpetuate prejudices across various applications. Literature studies have demonstrated that Machine Learning systems often reflect and amplify social biases due to many factors, like unbalanced or underrepresented data, training strategies, and model assumptions Mehrabi et al. (2021); Zhou et al. (2024).

Silva (2020) presented a study about *online racism*, where the author mapped some cases of Algorithmic Racism related to Machine Learning and search systems, and Social Media algorithms. Obermeyer et al. (2019) presented examples of how racial bias in machine learning models affects critical domains such as healthcare. The study found that, at the same risk score, Black patients are significantly sicker than White patients due to signs of uncontrolled illnesses and the algorithm bias from using healthcare costs as a proxy for illness, which disadvantages Black patients due to unequal access to care. The findings emphasize that addressing racial bias in machine learning requires technical interventions and a deep understanding of the sociocultural contexts in which these systems operate. Buolamwini & Gebru (2018) presented an approach to evaluate bias in automated facial analysis algorithms and datasets concerning phenotypic subgroups and brought critical attention to the racial biases embedded within machine learning systems. Their study demonstrated significantly higher error rates for darker-skinned individuals, particularly women, than lighter-skinned individuals. These findings highlighted the racial biases present in training datasets, where the under-representation of minority groups perpetuates inequities in algorithmic decision-making.

Recent studies have increasingly focused on uncovering social biases in multimodal models. Wolfe & Caliskan (2022) examines racial bias in vision-language systems and shows that these models tend to disproportionately associate the term "American" with white individuals, while often excluding or misrepresenting people of color in similar national identity contexts, thus reinforcing harmful racial stereotypes. Ghate et al. (2024) investigates gender bias in multilingual multimodal models, illustrating how these existing vision-language models reflect and amplify gendered cultural norms and fail to generalize fairly across languages, regions, and gender identities. Wang et al. (2022) investigates multilingual fairness in pre-trained multimodal architectures, emphasizing how these models perform inconsistently when representing different demographic and linguistic groups. Their work highlights disparities related to gender and geographic region in model outputs.

Currie et al. (2024), in their study, evaluated gender and ethnicity bias in generative AI text-to-image models, revealing biases in generative AI systems tasked to creating visual representations of Australian pharmacists. Using DALL-E 3 OpenAI (2024), the researchers analyzed how gender and ethnicity were represented in the generated images. The authors found a significant over-representation of White male pharmacists, with limited representations of women and individuals from minority racial or ethnic groups, which highlights the limitations of current generative models in reflecting real-world diversity. Park (2024) discussed how generative AIs reproduce biases and stereotypes against certain groups of people, which can exacerbate social inequities.

In the present work, our goal is, similar to the works mentioned above, to evaluate racial bias present in SOTA Captioning Models concerning race diversity by explicit description of race or implicit normalization of race White (Caucasian) and to reinforce this racial debate and bring more elements to contribute to this discussion about how White is not interpreted as a racial concept and how non-Whites races are made explicit by A.I. models.

## 3    MEASURING RACIAL BIAS IN CAPTIONS

### 3.1    DATASETS

Examining racial presence terms in captions requires analysis of the defined racial "classes". In this work, we use the racial class labels in two benchmark databases and the races presented in the Brazilian domain defined by Instituto Brasileiro de Geografia e Estatística (IBGE).

The **FairFace** dataset Karkkainen & Joo (2021) is a large-scale, diverse dataset designed to address biases in facial recognition and image analysis tasks. The dataset consists of over $96,000$ of human faces, with attributes annotations such as race, gender, and age. The FairFace prioritizes balanced representation across seven racial categories – `White`, `Black`, `Indian`, `East Asian`, `Southeast Asian`, `Middle Eastern`, and `Latino`. The **UTKFace** dataset Zhang et al. (2017) is a widely used facial image dataset developed for age, gender, and race-related research in machine learning. It contains over $20,000$ facial images, each annotated with age, gender, and one of five racial categories: `White`, `Black`, `Asian`, `Indian`, and "Others" (like `Hispanic`, `Latino`, `Middle Eastern`). Although the FairFace dataset has split annotations for Asian (*East & Southeast Asian*) and for *Middle Eastern & Latino*, we group these groups to follow the UTKFace's structure.

The **TSE** (in Portuguese, Tribunal Superior Eleitoral) dataset Tribunal Superior Eleitoral (2024) is a publicly available dataset provided by Brazil's Superior Electoral Court, containing extensive electoral and political data. This dataset includes candidate profiles (including photos), political affiliations, and government proposals. It is widely used in research on political science, election transparency, and bias detection in electoral processes Jacintho et al. (2020); Vasconcelos et al. (2022). The races present on the TSE dataset are `Branca`, `Parda`, `Preta`, `Indígena`, and `Amarela`- translated as White, Mixed-race or Brown, Black, Indigenous and yellow[1], this last one refers to individuals of Asian descent. Due to the large amount of historical data available from the latest released candidate dataset (Candidates - 2024), which has more than $460,000$ images, we selected 10% of samples stratified by race and gender.

## 3.2 METRICS

Although quantitative metrics are not our main focus or object of study, we proposed some metrics to better understand the impact of these models in terms of racialization. First, let's define the random variables $R$ as the race of an individual shown in the photo, $V$ as the race seen by the model. In addition, we defined $TP_r$ as the number of samples that are identified as race $r$ and are in fact of this race, $TN_r$ that are not identified as being of race $r$ and are in fact not of this race ($FP_r$ and $FN_r$ are defined similarly).

*Racialization Proportion* ($PR_r$). This proportion aims to evaluate how many samples are racialized for race $r$ ($TP_r + FP_r$) proportionally to the number of samples in the database.

$$PR_r \leftarrow \frac{TP_r + FP_r}{|R|} = \frac{P(V = r)}{P(R)} \tag{1}$$

*Correct Racial Attribution Index ($IRC_r$).* Aims to measure how many times the captions correctly identified the race $r$.

$$IRC_r \leftarrow \frac{TP_r}{TP_r + FN_r} = P(V = r | R = r)$$

*Incorrect Racial Attribution Index ($IRI_r$).* Measures how many times the captions identified another race $j$ but it was actually race $r$.

$$IRI_r \leftarrow \frac{\sum_{j \neq r} FP_j}{TP_r + FN_r} = \sum_{j \neq r} P(V = j | R = r)$$

*No Racial or Ethnic Attribution Reference ($NRE_r$).* Measures the proportion of samples whose captions have no racial attribution, in relation to the total occurrences of that race.

---

[1]The term yellow is considered outdated or inappropriate in many English-speaking contexts, where "Asian" is the preferred terminology.

$$NRE_r \leftarrow 1 - \frac{TP_r + \sum_{j \neq r} FP_j}{TP_r + FN_r} = 1 - P(V = r|R = r) - \sum_{j \neq r} P(V = j|R = r)$$

It is worth noting that $IRC_r + IRI_r + SRE_r = 1$ and these indices measure racialization within the sample set of race $r$. $PR_r$ measures the racialization index of the general population.

*Contextualizing the proposed metrics* Our proposed metrics can be interpreted through the lens of established fairness metrics:

The Racialization Proportion $PR_r$ captures the proportion that the model assigns a racial attribute $r$, independently of the true race. From this perspective, disparities in $PR_r$ across groups reflect differences in the likelihood of racialization and can be interpreted as a form of Demographic Parity violation, where certain racial attributes are disproportionately assigned.

Furthermore, with metrics $IRC_r$ and $IRI_r$ we can compare the rates at which individuals of different racial groups are correctly (or incorrectly) assigned a given racial attribute. If the proportion is the same across groups when the individual truly belongs to that race, the model exhibits Equality of Opportunity: individuals of race (r) have an equal chance of being correctly (or incorrectly) identified as such, regardless of other factors.

Finally, $NRE_r$ measures the absence of racial attribution, enabling a complete characterization of model behavior for each group.

### 3.3 CAPTION EVALUATION SYSTEM – RACIAL DISCRIMINATION

**Image Captioning.** Our primary focus in this work is to measure racial presence-absence terms in sentence-generated captions. For the image captioning task, in this work, we utilized two models: InstructBLIP Dai et al. (2023) and Llama 3.2-Vision AI @ Meta Llama Team (2024). The Instruct-BLIP (Salesforce/instructblip-vicuna-7b), is a visual instruction-tuned variant of the BLIP-2 Li et al. (2023) model that uses the Vicuna-7b Zheng et al. (2023) language model as its backbone which was fine-tuned on a mixture of chat and instruct datasets. The Llama 3.2-Vision (meta-llama/Llama-3.2-11B-Vision-Instruct) is an optimized collection for visual recognition, image reasoning, captioning, and answering general image questions. Both models are designed to generate human-like textual descriptions and were chosen for their capability of effectively understanding and generating language in the context of visual information.

*Prompt Diversity.* We provided each model with a tuple consisting of an image and a corresponding instruction. We utilized a set of 13 distinct image captioning instruction prompts proposed by Dai et al. (2023) (A detailed list is provided in Appendix B), randomly selecting one instruction per image.

**Race Identification in Caption.** Following the caption generation step, we employed a Large Language Model (LLM) to detect the presence of racial terms in the generated captions. Specifically, we used the Llama 3-Text (Meta-Llama-3-8B-Instruct) model, a pretrained and instruction-tuned generative text model. The LLM model receives an instructional prompt and a generated caption and determines whether the caption includes explicit racial references (The Figure 2 in Appendix A illustrates this process).

*Judge Evaluation* To evaluate the effectiveness of this model in identifying racialized content, we constructed a custom dataset of 1000 manually verified non-racialized captions (i.e., captions containing no references such as "Black man", "White person", or "Indian woman"). We then created an additional 1000 captions by artificially inserting racial adjectives into the original non-racialized samples. This resulted in a balanced dataset of 2000 captions, with 50% containing racial terms and 50% not. We conducted an experiment using a LLM to identify whether the caption includes explicit racial references (following the structure shown in Appendix A), recording the model's prediction for each of the 2000 samples. The Llama 3-Text achieved an overall accuracy of 93.2% in correctly identifying the presence or absence of racial terms. When disaggregated by group, we observed slight variations in performance: the model's lowest accuracy was 88% for

the term *negro* (a term commonly used in Brazil similarly to 'black' in English), followed by $91\%$ for *white*. The best-performing categories include *African American* ($98.8\%$), *Black* ($98.6\%$), and *Asian* ($97.7\%$). Also, the model achieves an accuracy of $89.2\%$ for the *No race or ethnicity* category, suggesting occasional false positives, where it incorrectly infers racial content in neutral captions.

Moreover, although this validation dataset is partially artificially constructed through controlled insertion of racial descriptors, we ensure coverage of both English and Portuguese terms, reducing the likelihood of distributional mismatch with the evaluation setting. While incorporating additional validation mechanisms, such as rule-based extractors or alternative judge models, could further strengthen robustness, we believe these results suggest that the LLM judge is effective across a diverse set of racial descriptors, including both English and Portuguese terms. Given the high overall accuracy and the absence of large systematic disparities, we expect that any residual errors are unlikely to meaningfully affect the conclusions of our downstream fairness analysis.

## 3.4 RESULTS AND DISCUSSIONS IN RACIAL BENCHMARKS DATABASES

To assess the measurement of racial presence vs. absence in the generated captions following the proposed pipeline, we employ the metrics described in Section 3.2. Table 1 presents the Racialization Proportion ($PR_r$), in which we can observe a significantly higher proportion for the `Black` and `Asian` races across different models and datasets when compared to the other races. The `Indian` race presents higher values with the InstructBLIP model than with Llama 3.2-Vision in both datasets. For `Latino-Hispanic` & `Middle Eastern` (LH & ME), this value is considerably lower than for the other groups across both models and datasets, indicating that the models do not generate direct or explicit references to these groups. Similarly, the behavior of the InstructBLIP model for the `White` race suggests that this race is described less frequently than other groups, and although Llama 3.2-Vision produces a reasonable proportion in the FairFace dataset, it still remains lower than the proportions observed for `Black` and `Asian`, for example.

| **MM-LLM** | **Dataset** | **White** | **Black** | **Asian** | **Indian** | **LH & ME** |
|---|---|---|---|---|---|---|
| Reference Values | UTKFace | 0.424 | 0.189 | 0.149 | 0.167 | 0.071 |
| | FairFace | 0.191 | 0.141 | 0.267 | 0.142 | 0.260 |
| InstructBLIP | UTKFace | 0.001 | 0.033 | 0.033 | 0.028 | 0.000 |
| | FairFace | 0.001 | 0.015 | 0.056 | 0.013 | 0.001 |
| Llama-3.2-Vision | UTKFace | 0.026 | 0.046 | 0.015 | 0.018 | 0.001 |
| | FairFace | 0.030 | 0.072 | 0.036 | 0.006 | 0.001 |

Table 1: Results of the Racialization Proportion ($PR_r$), measuring for each race the proportion of samples containing racial mentions in the generated sentences relative to the number of samples in the dataset. Higher values of this metric indicate a greater occurrence of racialized sentences. `Latino-Hispanic-Middle Eastern` present the lowest values, suggesting that these races are less explicitly mentioned in the captions. The **Reference values** denote the actual proportion of samples of each race in the datasets.

We believe that, in an symmetric world, these proportions should be similar across all groups. This brings us back to the concept of equalized odds for fairness Hardt et al. (2016), which requires that a system maintain comparable false positive and false negative rates across groups. To conduct a phenotypic performance analysis and formalize this notion of fairness in our evaluated captioning models, the differences for each race in each model are compared in Table 2. For each race, we report the percentage of sentences in which race is predicted correctly ($IRC_r$) or incorrectly ($IRI_r$) and when no racial term is generated ($NRE_r$). Across all models, the results for `Black-Asian-Indian` stand out. Interestingly, we observe that, mainly on InstructBLIP, the `White` has minimal values of corrects ($IRC_r$) and incorrects ($IRI_r$), with higher values for ($NRE_r$), which indicates that the model does not generate captions with explicit racial reference, preferring the generation of texts without racialized content for this group. However, while the results for `Latino-Hispanic` & `Middle Eastern` also present small ($IRC_r$) values, the

($IRI_r$) results on both models indicate, at some moments, the generation of texts with racialized content.

| | | InstructBLIP | | | Llama 3.2-Vision | | |
|---|---|---|---|---|---|---|---|
| | | $IRC_r$ | $IRI_r$ | $NRE_r$ | $IRC_r$ | $IRI_r$ | $NRE_r$ |
| **White** | **UTKFace** | 0.000 | 0.007 | 0.992 | 0.047 | 0.009 | 0.943 |
| | **FairFace** | 0.001 | 0.003 | 0.995 | 0.078 | 0.010 | 0.911 |
| **Black** | **UTKFace** | 0.166 | **0.105** | 0.728 | **0.204** | 0.046 | 0.749 |
| | **FairFace** | 0.095 | **0.087** | 0.816 | **0.304** | 0.049 | 0.645 |
| **Asian** | **UTKFace** | **0.211** | 0.023 | 0.764 | 0.090 | 0.033 | 0.875 |
| | **FairFace** | **0.195** | 0.012 | 0.792 | 0.109 | 0.042 | 0.847 |
| **Indian** | **UTKFace** | 0.152 | 0.010 | 0.836 | 0.101 | 0.041 | 0.856 |
| | **FairFace** | 0.078 | 0.018 | 0.902 | 0.038 | **0.150** | 0.811 |
| **LH & ME** | **UTKFace** | 0.001 | 0.031 | 0.967 | 0.004 | **0.052** | 0.943 |
| | **FairFace** | 0.001 | 0.016 | 0.982 | 0.002 | 0.072 | 0.924 |

Table 2: Results of the racialization indices by race,model and dataset combination. We highlight the highest values (**in bold**) and the lowest values (underlined) of the metrics.

**Qualitative Analysis.** We analyzed the captions generated by both models to evaluate the linguistic manifestation of asymmetric racialization. While InstructBLIP produces concise, direct captions and Llama 3.2-Vision generates elaborate descriptions, both exhibit a systematic disparity in racial attribution. Notably, despite its brevity, InstructBLIP consistently identifies non-White individuals through explicit markers (e.g., 'Black man' or 'Asian woman') while omitting racial descriptors for White subjects (or when the individual 'appears' white), even when phenotypic cues like 'blonde' are present. This confirms that racial 'marking' is a structural priority for non-White subjects even under linguistic constraints. Conversely, Llama 3.2-Vision's detailed nature allows for more frequent phenotype-specific terms such as 'fair-skinned,' 'dark-skinned', or 'Caucasian'; however, this increased detail often leads to incorrect projections of Western racial categories onto diverse subjects.

Together, these behaviors on both models reinforce 'Whiteness as Default' by treating White phenotypes as the universal, unmarked norm while explicitly racializing others. Historically, Whiteness has been constructed as colorless and positioned as "the human norm" Knight (2006); Cardoso (2010), often described in universal terms as representing "people" in general Dyer's (1997); Knight (2006). Further corroborated by Table 1 and Table 2, the presented results suggest that this notion is perpetuated and reinforced by the InstructBLIP model.

## 3.5 RESULTS AND DISCUSSIONS IN BRAZILIAN SUPERIOR ELECTORAL COURT TSE DATASET

We extended our analysis of racialized references to the TSE-Brazil dataset to evaluate model alignment with localized racial constructs. We observed a significant discrepancy between the model's generated terminology and official Brazilian categories (Branca, Preta, Parda, Amarela, and Indígena). To facilitate a comparative analysis, we systematically mapped generated terms to these official labels: White and *Caucasian* were associated with Branca; Black, *African*, and *African-American* with Preta; *Asian* (including Asian-American) with Amarela; and *Indigenous* or *Native American* with Indígena. Notably, as explicit references to *Brown* or *Mixed-race* were scarce in both models, we included *Latin* and *South American* descriptors within the *parda* category. This alignment gap highlights the limitations of Vision-Language Models in adapting to racial classifications outside the Global North.

Table 3 presents the results of the Racialization Proportion ($PR_r$) after this process. We can observe that LLaMA 3.2-Vision exhibits a significant proportion for race Amarela, remaining very close to the reference value for this race. Both models demonstrate a similar tendency to assign low or no percentages to the Branca and Parda races. The Preta race shows a significant proportion relative to the reference value in both models, while the Indígena race reaches the lowest values.

| MM-LLM | Branca | Preta | Parda | Indígena | Amarela |
|---|---|---|---|---|---|
| Reference Values | 0.471 | 0.114 | 0.405 | 0.006 | 0.004 |
| InstructBLIP | 0.000 | 0.026 | 0.000 | 0.000 | 0.002 |
| Llama-3.2-Vision | 0.010 | 0.016 | 0.002 | 0.000 | 0.003 |

Table 3: Results of the Racialization Proportion ($PR_r$), measuring for each race the proportion of samples containing racial mentions in the generated sentences relative to the number of samples in the TSE dataset.

We analyzed the accuracy of racial attributions within the TSE dataset (Table 4) to evaluate how these models align with localized racial constructs. While `Preta` category a consistent highest correct scores ($IRC_r$) and lowest values of $NRE_r$; and *branca* produced consistent highest omission rate ($NRE_r$), reinforcing the black racialization and 'Whiteness as Default' behavior even in a localized context. However, *amarela* and *indígena* individuals were incorrectly assigned to other categories. This reveals a fundamental misalignment: the racial stereotypes embedded in these models fail to capture the complexity of Brazilian racial diversity.

| | MM-LLM | $IRC_r$ | $IRI_r$ | $NRE_r$ |
|---|---|---|---|---|
| **Branca** | **InstructBLIP** | 0.000 | 0.011 | 0.989 |
| | **Llama-3.2-Vision** | 0.017 | 0.007 | 0.975 |
| **Preta** | **InstructBLIP** | 0.119 | 0.010 | 0.871 |
| | **Llama-3.2-Vision** | 0.097 | 0.010 | 0.892 |
| **Parda** | **InstructBLIP** | 0.000 | 0.027 | 0.973 |
| | **Llama-3.2-Vision** | 0.002 | 0.025 | 0.971 |
| **Amarela** | **InstructBLIP** | 0.017 | 0.017 | 0.967 |
| | **Llama-3.2-Vision** | 0.044 | 0.016 | 0.938 |
| **Indígena** | **InstructBLIP** | 0.012 | 0.023 | 0.965 |
| | **Llama-3.2-Vision** | 0.000 | 0.042 | 0.957 |

Table 4: Results of the racialization indices by race and models in the TSE dataset.

The notably low racialization of *Pardo* individuals suggests a process of 'digital whitening', where this group is often homogenized into the *branca* category. This confusion underscores the model's tendency to oversimplify complex identities and fail to generalize across contexts of colorism. In InstructBLIP, the lack of explicit 'confusion' with the White category is merely a byproduct of the model's total failure to racialize White subjects in general (as can be seen in Table 1, and Table 3).

Collectively, these findings support our hypothesis that models reproduce a whitening process that erases the specificity of mixed-race identities in the Brazilian context. We argue that the primary harm is the asymmetry in which one race is the "unmarked norm" and others are "racialized". Whether a model should always or never include race is a design decision; however, it must be equitable. Current "digital whitening" is an unacceptable erasure of black origins in Pardo identity.

**Embedding Experiment.** To investigate the structural roots of the observed oversimplification bias, we analyzed the embedding space of the InstructBLIP vision encoder across all three datasets. We computed group-level membership probabilities to assess how racial groups are clustered and separated in the latent space. This approach converts individual cosine distances into aggregate likelihoods, yielding a normalized probability distribution that reflects the representational alignment of each category. A detailed mathematical derivation of this group membership metric and our $k$-nearest neighbor density estimation approach is provided in Appendix C.

We used the FairFace dataset as a reference to evaluate the structural roots of oversimplification bias. Initial comparisons between UTKFace and FairFace embeddings (Figure 1(a)) show coherent alignment, with images clustering according to their depicted racial groups. However, Figure 1(b) reveals a significant failure in contextual adaptability for the TSE-Brazil dataset. The model predominantly collapses Brazilian racial categories into a generic `Latino_Hispanic` cluster, with only the `Preta` race showing a distinct alignment toward the `Black` category.

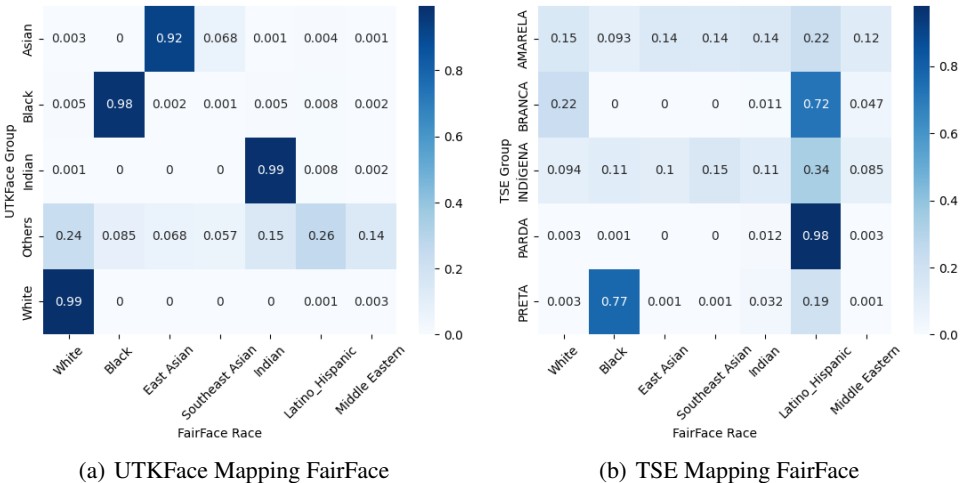

(a) UTKFace Mapping FairFace       (b) TSE Mapping FairFace

Figure 1: Cosine similarity analysis of image embeddings extracted from the InstructBLIP vision encoder across datasets. UTKFace shows race-consistent alignment with FairFace, while TSE embeddings are oversimplified and predominantly mapped to `Latino_Hispanic`

In a posterior analysis, when excluding the `Latino_Hispanic` group to isolate more granular associations, we observe a critical representational harm in the TSE dataset: the embeddings of `Parda` individuals show a higher probability of similarity to the `White` category within the InstructBLIP vision encoder. This indicates a 'whitening bias' that exists at the latent level, prior to caption generation. This finding provides empirical evidence that the model's internal representations oversimplify and homogenize mixed-race populations, structurally reinforcing Whiteness as the default prototype even for diverse Brazilian phenotypes.

## 4 LIMITATIONS

While this study provides a robust analysis of racialization in VLMs, several limitations define its scope. First, we utilize foundation models (InstructBLIP and Llama-3.2-Vision) that, while not the most recent iterations, remain the dominant Global North architectures currently exported to and utilized within the Global South. Analyzing their "out-of-the-box" behavior is critical for understanding the systemic biases embedded in real-world applications today. Second, we adopt a single-axis focus on race; we recognize that intersectionality (the compounding effects of gender and age) is essential for a complete understanding of representational harm, but it remains out of scope for this specific work.

Linguistically, our evaluation was conducted in English to standardize comparisons across global and local datasets. However, using a Global North "Judge" (Llama-3) to evaluate Brazilian racial constructs introduces potential cultural constraints. We acknowledge that terms such as Global North and Global South are simplifications used to facilitate discursive analysis of power dynamics rather than absolute geographical or economic definitions. Furthermore, sociopolitical terms such as whitening (branqueamento), miscegenation (miscigenação), and colorism (colorismo) carry immense historical and analytical weight in Brazil. In this study, these terms are employed as discursive frameworks. They should be understood within this specific technical and sociological context, representing established categories of analysis in Brazilian racial studies rather than biological descriptions.

## 5 CONCLUSION

This paper introduced an automated pipeline to quantify racial attribution disparities in SOTA Vision-Language Models (VLMs), specifically InstructBLIP and Llama 3.2-Vision. By evaluating these models across global benchmarks (UTKFace, FairFace) and a localized Brazilian dataset (TSE), we provide empirical evidence of how "Whiteness as Default" is codified in multimodal architectures.

Our findings confirm that VLMs exhibit a significant asymmetric racialization and bias persists and intensifies in racially mixed societies, where models fail to align with localized racial constructs. Specifically, when confronted with the Brazilian `Parda`, `Indígena` and `Amarela` categories-identities uncommon in Global North training sets-the models default to oversimplified or Western-centric categories.

Furthermore, our latent space analysis reveals the potential representational origins of this bias. We demonstrate that the "whitening" observed in textual captions is also rooted in the vision encoder's embedding space, which mathematically clusters mixed-race phenotypes with White prototypes. This "digital whitening" erases the specificity of Brazilian racial diversity, effectively automating historical processes of cultural domination and "branqueamento" (whitening).

By identifying these structural biases, this study highlights the urgent need for bias mitigation that moves beyond "one-size-fits-all" fairness metrics. Preserving these algorithmic structures risks technicalizing historical privileges and undermining democratic and racial achievements. Future work must focus on decentralizing Global North racial logic to ensure that generative AI does not become an engine for representational erasure in the Global South.

## GENERATIVE AI DISCLOSURE STATEMENT

Generative AI tools (ChatGPT, version GPT-5.2) were utilized exclusively for grammatical correction, stylistic enhancements, and lexical refinements to ensure linguistic clarity. The formulation of research questions, experimental design, embedding-space analysis, and all scientific conclusions regarding racial bias and the Brazilian context remain entirely the original work of the authors.

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

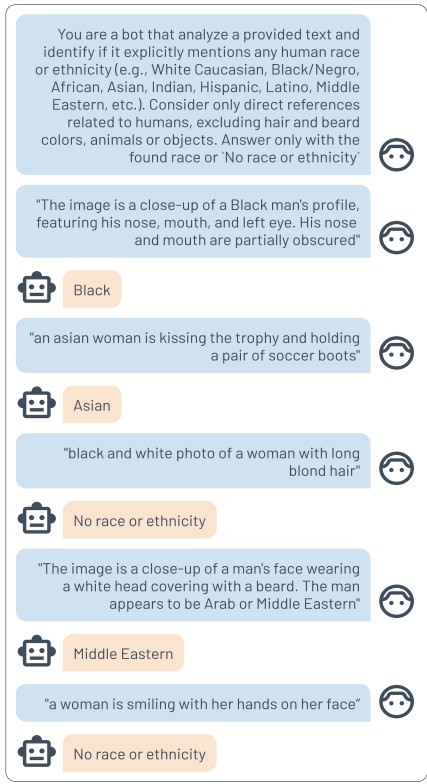

Figure 2: A few examples of racial term presence predictions generated by Llama 3-Text model. The first row displays the instructional prompt used, followed by the generated captions and the corresponding answer from the model.

## A   LLM AS A JUDGE EXAMPLE

## B   INSTRUCTION PROMPTS FOR IMAGE CAPTIONING

| | Instruction Prompt Templates |
|---|---|
| 1 | <Image>A short image caption: |
| 2 | <Image>A short image description: |
| 3 | <Image>A photo of |
| 4 | <Image>An image that shows |
| 5 | <Image>Write a short description for the image. |
| 6 | <Image>Write a description for the photo. |
| 7 | <Image>Provide a description of what is presented in the photo. |
| 8 | <Image>Briefly describe the content of the image. |
| 9 | <Image>Can you briefly explain what you see in the image? |
| 10 | <Image>Could you use a few words to describe what you perceive in the photo? |
| 11 | <Image>Please provide a short depiction of the picture. |
| 12 | <Image>Using language, provide a short account of the image. |
| 13 | <Image>Use a few words to illustrate what is happening in the picture. |

Table 5: Instruction prompts used for the captioning task. These prompts were proposed by Dai et al. (2023).

## C  GROUP MEMBERSHIP PROBABILITY

This formulation addresses the scenario where a dataset of samples, is assumed to have been generated entirely by one specific component from a mixture of possible distributions. We aim to calculate the posterior probability that the latent component index corresponds to distribution , given the entire observed set .

**Mathematical Formulation**  Using Bayes' Theorem, the posterior probability $P(z = j|X)$ is defined as:

$$P(z = j|X) = \frac{P(X|z = j)P(z = j)}{\sum_{l=1}^{k} P(X|z = l)P(z = l)}$$

where $P(z = j)$ is the prior probability (mixing weight) of component $j$, denoted as $\pi_j$; $P(X|z = j)$ is the likelihood of the entire group $X$ assuming it originated from component $j$.

Under the assumption that the samples are independent and identically distributed (i.i.d.) given the component, the group likelihood is the product of the individual probability density functions (PDFs):

$$P(X|z = j) = \prod_{i=1}^{n} f_j(x_i)$$

Substituting this likelihood back into the posterior equation yields the final probability for the group's membership:

$$P(z = j|X) = \frac{\pi_j \prod_{i=1}^{n} f_j(x_i)}{\sum_{l=1}^{k} \left( \pi_l \prod_{i=1}^{n} f_l(x_i) \right)}$$

This metric quantifies how well distribution $j$ explains the aggregate behavior of the dataset $X$ relative to the other available distributions in the mixture. Because the likelihoods are multiplicative, the resulting probability reflects the accumulated evidence from all $n$ samples simultaneously.

**Implementation.**  In order to estimate $f(x)$ we used Loftsgaarden & Quesenberry (1965) approach that considers the distance to $k$-th nearest neighboor. The estimator is defined as:

$$\hat{f}_n(z) = \left( \frac{k(n) - 1}{n} \right) \frac{1}{A_{r_{k(n)}, z}}$$

where $n$ is the sample size, $k(n)$ is an integer parameter set to approximately $\sqrt{n}$, and $A_{r_{k(n)}, z}$ represents the volume of the hypersphere determined by the distance $r_{k(n)}$ to the $k$-th nearest neighbor. While the original formulation uses Euclidean distance, the authors state that other metrics are equally valid; therefore, our implementation utilizes the cosine distance.

