# OpenReview forum: "When AI Describes Race? Unveiling Racial Bias in Vision-Language Models in Brazilian People"
_ICLR.cc/2026/Workshop/AFAA — AFAA 2026 Oral_

### Official Review · Reviewer_azSX · 2026-02-20

**Rating:** 4
**Confidence:** 4

**Summary:**

This work utilizes a new image dataset of faces from Brazil to further analyses on disparities by race/skintone. It identifies notable performance gap in several models, suggesting areas for future improvements.

**Strengths:**

+ The work includes an interesting analysis using the Brazilian Superior Electoral Court TSE dataset, which provides an additional insight of race/skin-tone related biases in VLMs and allows for additional subcategories, like mixed-race.
+ The paper studies usefully explores potential root causes of observed model discrepancies by exploring the embeddings of the evaluation datasets and similarities between different subcategories.
+ The work is usefully motivated with extensive social science literature and specific contextualization of race / skintone classifications in Brazil.

**Weaknesses:**

- The methodological novelty of the UTKFace/FairFace analysis is somewhat limited and the findings with default bias towards Caucasian appearances and poorer performance for non-white images is aligned with  much of recent fairness literature, suggesting limited novelty.
- The models studied are a bit dated at this point; More recent and ubiquitous models could be useful to include as well.

---

### Official Review · Reviewer_hXck · 2026-02-20
**The paper studies how vision-language models mark race in image captions and presents consistent results across datasets. While the work is clear and relevant, it would benefit from a clearer fairness definition and stronger methodological support.**

**Rating:** 4
**Confidence:** 4

**Summary:**

The paper studies how vision-language captioning models mention race in image descriptions and evaluates model fairness based on whether some racial groups are marked more often than others. It evaluates two models on three datasets, including a Brazilian dataset, and analyzes both the generated captions and the underlying embeddings. The results show consistent patterns across models and datasets, and the discussion places these findings in a broader social context.

**Strengths:**

1. The use of Brazilian racial categories allows the authors to compare model behavior across global benchmarks and a local dataset. This helps highlight how models trained primarily in Global North contexts may not align well with Global South racial taxonomies.

2. The paper connects its empirical findings to broader social discussions, including the concept of “Whiteness as Default” and Brazilian processes such as whitening and colorism.

3. The analysis goes beyond surface captions by examining embedding representations. This supports the argument that disparities may arise not only in caption generation but also in internal vision encoder representations.

4. Although the evaluation relies on an LLM judge, the authors partially address this concern by validating its performance on a manually verified dataset and reporting its accuracy.

**Weaknesses:**

1. The paper would benefit from further clarifying why differences in explicit racial mention should be interpreted as harm. In particular, it would help to explain why equal mention rates across groups are considered the appropriate fairness objective in this setting.

2. The proposed Racialization Proportion (PRr) is conceptually related to demographic parity. Making this connection more explicit and reporting results in terms of established fairness metrics (e.g., demographic parity gaps or equalized odds) would strengthen the theoretical grounding and improve interpretability.

3. Differences in image quality, lighting, facial expression, as well as gender and age, may affect how race is mentioned. Considering these factors could strengthen the analysis.

4. Since the evaluation relies on an LLM judge, adding a robustness check (e.g., a rule-based extractor or another judge model) would increase confidence in the results.

---

### Official Review · Reviewer_ww8V · 2026-02-21
**A very focused race flattening analysis pipeline for brazillian racial groups**

**Rating:** 4
**Confidence:** 3

**Summary:**

This paper tries to provide a pipeline where they analyze how the brazillian racial categories are flattered or "digitally whitened". Overall, their pipeline uses the fairface and utk dataset and produce image captions. he paper reports disparities in explicit racial mentions and argues that Brazilian racial categories (e.g., mixed-race identities) are often flattened or “digitally whitened,” showing a mismatch between localized constructs and the model behavior.

**Strengths:**

- Judge evaluation: The authors have created a human validated set for race identification in image captions, adding confidence.
- A nice idea to validate suppression of other non-dominant racial categories/terms
- Whitening analysis is a good contribution, specifying how Brazilian racial diversity is being erased

**Weaknesses:**

- Using only 2 models for image captioning is insufficient, maybe add a couple more models
- Using only english as the language. More native language could yield different results, and it is important to capture them eventually.
- Using LLM judge can still introduce inaccurate results as the racial terms in the validation set may not reflect the test set

---

### Meta-Review · Area_Chair_7gcq · 2026-02-26

**Recommendation:** Main Papers Track
**Confidence:** 5

**Metareview:**

The paper presents an analysis pipeline for how vision-language captioning models flatten Brazilian racial categories (including mixed-race identities) and exhibit patterns consistent with “whiteness as default,” using caption outputs and embedding-space analyses across multiple datasets (including a Brazilian dataset). Reviewers value the strong contextual grounding in Brazilian racial taxonomies and social science, the “digital whitening” framing, and efforts to increase confidence via a human-validated set for race identification in captions alongside embedding analyses that probe potential root causes. The main concerns are about scope and robustness: limited and somewhat dated model coverage, English-only evaluation, and reliance on an LLM judge (suggesting broader models and additional validation/robustness checks). Overall, the reviews are positive and I recommend acceptance.

---

### Decision · Program_Chairs · 2026-03-02

Accept (Oral)